# Polypeptides derived from α-Synuclein binding partners to prevent α-Synuclein fibrils interaction with and take-up by cells

**Elodie Monsellier, Maya Bendifallah, Virginie Redeker, Ronald Melki** *

CEA, Institut François Jacob (MIRcen) and CNRS, Laboratory of Neurodegenerative Diseases (UMR9199), Fontenay-aux-Roses, France

* ronald.melki@cnrs.fr

**Data Availability Statement:** All relevant data are within the manuscript and its Supporting Information files.

## Abstract

α-Synuclein (αSyn) fibrils spread from one neuronal cell to another. This prion-like phenomenon is believed to contribute to the progression of the pathology in Parkinson's disease and other synucleinopathies. The binding of αSyn fibrils originating from affected cells to the plasma membrane of naïve cells is key in their prion-like propagation propensity. To interfere with this process, we designed polypeptides derived from proteins we previously showed to interact with αSyn fibrils, namely the molecular chaperone Hsc70 and the sodium/potassium pump NaK-ATPase and assessed their capacity to bind αSyn fibrils and/or interfere with their take-up by cells of neuronal origin. We demonstrate here that polypeptides that coat αSyn fibrils surfaces in such a way that they are changed affect αSyn fibrils binding to the plasma membrane components and/or their take-up by cells. Altogether our observations suggest that the rationale design of αSyn fibrils polypeptide binders that interfere with their propagation between neuronal cells holds therapeutic potential.

## Introduction

The aggregation of proteins into fibrillar high molecular-weight species is involved in human degenerative diseases, including Alzheimer's, Parkinson's, or Huntington's [1]. During the last decade, it has become evident that those protein aggregates traffic between neuronal cells and amplify by seeding the aggregation of their constituting proteins [2–5]. This prion-like phenomenon is thought to be responsible for the stereotypic progression of the pathology in the brain [2,5]. Impeding this phenomenon would be valuable to slow down the progression of disease [6,7].

The spread of amyloid fibrils is a vicious circle involving different steps. First, protein aggregates form with time within neuronal cells [8]. They next escape actively, through export, or passively, upon cell death, the cells where they form [9–12]. The extracellular aggregates dock next to the membrane of naïve neuronal cells [13,14]. This membrane binding steps is followed by the internalization of the fibrils, mainly through endocytosis [15,16]. Once in the cells the aggregates reach the cytoplasm, where they recruit the otherwise soluble endogenous protein they are made of [17], after compromising endo-lysosomal integrity [18].

**Funding:** MB, Région Ile de France through DIM Cerveau et Pensée; RM, Institut de France-Fondation Simone et Cino Del Duca; RM, Fondation Pour La Recherche Médicale (contract DEQ. 20160334896); RM, EC Joint Programme on Neurodegenerative Diseases and Agence Nationale pour la Recherche (TransPathND, ANR-17-JPCD-0002-02 and Protest-70, ANR-17-JPCD-0005-01). Nothing to disclose. The funders had no role in study design, data collection and analysis, decision to publish, or preparation of the manuscript.

**Competing interests:** The authors have declared that no competing interests exist.

**Abbreviations:** αSyn, α-Synuclein; NKA, Na⁺/K⁺-ATPase; NBD, Nucleotide Binding Domain; SBD, Substrate Binding Domain; TEV, tobacco etch virus; ThT, Thioflavin T.

Alternatively, they imbalance neuronal proteostasis and trigger the de-novo aggregation of other aggregation prone proteins involved in neurodegenerative diseases [19]. The circle completes when amplified aggregates escape into the extracellular media, targeting new cells.

Every single step of the prion-like propagation process is a potential target for the development of new drugs that would delay the progression of disease. The binding of the extracellular aggregates to the membrane is especially attractive for different reasons [7]. As it takes place in the extracellular environment, it is more easily targetable than intracellular mechanisms [20]. Its underlying molecular mechanisms have been particularly well studied over the past few years [21]. The fibrils bind laterally to the plasma membrane [13]. The binding is mediated by interaction with the plasma membrane lipids [22], with different proteins partners [23–26] and with the extra cellular matrix components [27,28]. The efficiency of the binding depends both on the aggregates characteristics, such as their primary sequence [29,30], their net charge [22], their size [13] or their conformation [17], and on the properties of the membrane, with an emphasis on the role of the membrane curvature [31] and a specific lipid [32] and protein [24,25] composition.

As different membrane components are involved in the interaction with extracellular aggregates in their prion-like propagation process, it seems unlikely that targeting one of them would exert an effect. We therefore decided to target the fibrils themselves, coating them with peptide ligands so that their surface properties are changed and their interaction with membrane partners is compromised. We decided to develop polypeptide binders of fibrils as from a clinical point of view such binders are specific and safe, and their poor pharmacokinetics properties are amenable to optimization [33,34]. Incidentally over 60 peptide drugs have now reached the market [35]. Using a cross-linking and mass spectrometry strategy, we previously mapped the surface interfaces between αSyn monomers or fibrils and two protein partners, namely, the molecular chaperone Hsc70 [36–38] and the sodium/potassium pump Na+/K +-ATPase (NKA) [25]. Here, we designed a set of polypeptides based on the Hsc70 and NKA surface areas we identified to interact with αSyn. We assessed the interaction of the polypeptides derived from Hsc70 and NKA with fibrillar αSyn in vitro. We identify Hsc70-derived polypeptides that bind best αSyn fibrils. We also show that an NKA-derived peptide affect fibrils binding to Neuro-2a cells. Overall, our results lay the basis for developing further such polypeptides and improving their affinity for αSyn fibrils, so that their interactions with and uptake by Neuro-2a neuronal cells are affected.

## Results

### Hsc70 binds to αSyn fibrils with a high affinity, preventing their interaction with the plasma membrane and their take-up by cultured cells

We previously brought evidence for Hsc70 interaction with fibrillar αSyn using a sedimentation assay [36]. The dissociation constant we measured was 0.1 μM. Here we confirmed the interaction between Hsc70 and fibrillar αSyn using the same sedimentation assay followed by quantitative analysis of the proteins in the pellet and the supernatant fractions by SDS-PAGE (**Fig 1A**). Hsc70 alone remains in the supernatant, whereas it is pulled into the pellet when incubated for 1h at room temperature with pre-formed αSyn fibrils. To determine the affinity of Hsc70 for fibrillar αSyn we incubated preformed αSyn fibrils (1 μM) with increasing amount (0–2 μM) of a mix of unlabelled and ATTO488-labeled Hsc70 (labelled:unlabelled molar ratio of 1:50) for 1h at room temperature. The samples were then filtered through cellulose acetate membranes that retains fibrillar αSyn along with their binders, and the amount of ATTO488-Hsc70 was quantified by fluorescence measurements (**Fig 1B**). We measured a dissociation constant ($K_D$) between Hsc70-ATTO488 and αSyn fibrils of 0.49 ± 0.02 μM, consistent with previously published values [36,39]. We demonstrated that the binding between the

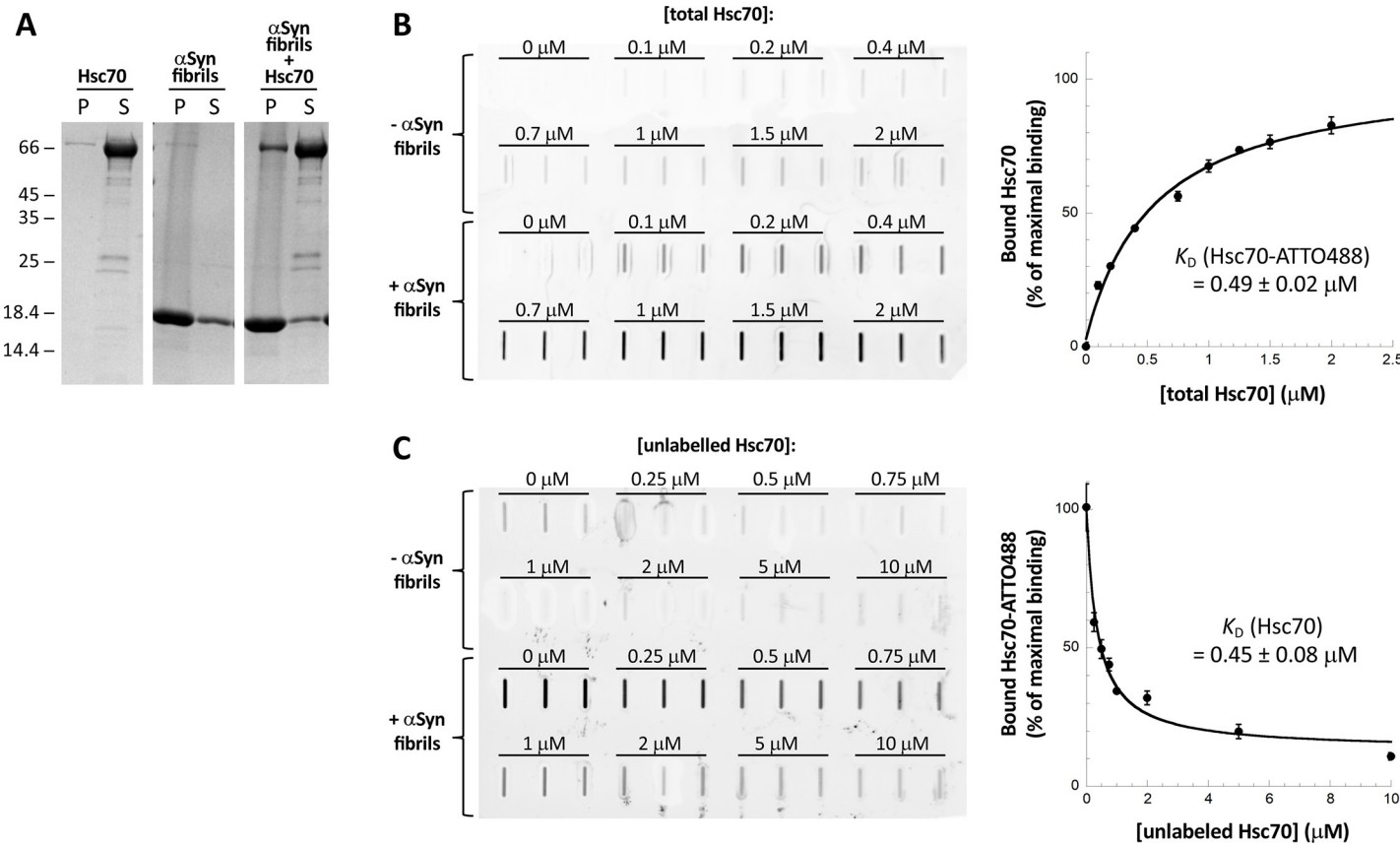

**Fig 1. Hsc70 binds to αSyn fibrils with high affinity. A,** Hsc70 binds to αSyn fibrils *in vitro*. SDS-PAGE analysis of the pellet (P) and supernatant (S) fractions of Hsc70 (10 μM), fibrillar αSyn (100 μM), and fibrillar αSyn (100 μM) incubated with Hsc70 (10 μM) for 1 h at RT. **B,** Quantification of Hsc70-ATTO488 binding to αSyn fibrils using the cellulose acetate filter trap assay. Hsc70-ATTO488 was diluted with unlabelled Hsc70 (labelled:unlabelled molar ratio of 1:50) to different final concentrations (0–2 μM) and incubated with or without αSyn fibrils (1 μM) for 1h at RT. Each sample was filtered in triplicate through a cellulose acetate membrane and the amount of Hsc70-ATTO488 trapped onto the membrane was quantified. The mean amount of Hsc70-ATTO488 bound to the αSyn fibrils normalized to the amount of Hsc70-ATTO488 bound at the maximal concentration used ("% of maximal binding") and the associated standard error values were calculated from 2 to 3 independent experiments. A filter trap membrane from one representative experiment is shown. **C,** Unlabeled Hsc70 compete with Hsc70-ATTO488 for binding to αSyn fibrils. A fixed concentration of Hsc70-ATTO488 (0.2 μM) was incubated with increasing concentrations of unlabeled Hsc70 (0–10 μM) and with or without αSyn fibrils (1 μM). Each sample was then filtered in triplicate through a cellulose acetate membrane. The mean amount of Hsc70-ATTO488 bound to the αSyn fibrils and the associated standard error values were calculated from these triplicates.

two partners was not affected by Hsc70 labelling. Indeed, unlabeled Hsc70 competed in a dose-dependent way with the binding of labeled Hsc70 to αSyn fibrils (**Fig 1C**), and the $K_D$ between Hsc70 and αSyn fibrils was identical to the $K_D$ between Hsc70-ATTO488 and αSyn fibrils (0.45 ± 0.08 μM).

We next assessed the consequences of Hsc70 interaction with αSyn fibrils on fibrils binding to the cell membrane and subsequent internalization. We set-up two different assays to assess separately αSyn fibrils binding and internalization (**Fig 2**). Preformed Alexa488-labeled αSyn fibrils (**S1 Fig**) bound to cultured Neuro-2a cells within 30 min incubation in a dose-dependent manner as assessed by quantification of fluorescent foci at cell membranes (**Figs 2A and S2A**). The addition of Trypan blue quenched all the fluorescence, indicating that the fibrils are located at the plasma membranes. We previously demonstrated that monomeric αSyn does not bind to cells in such a way [40,41]. This robust cellular binding assay was next used to monitor the effect Hsc70-fibrillar αSyn interaction on fibrils binding to Neuro-2a cells. αSyn fibrils (1 μM) were pre-incubated with increasing amounts of Hsc70 (0–10 μM). Neuro-2a

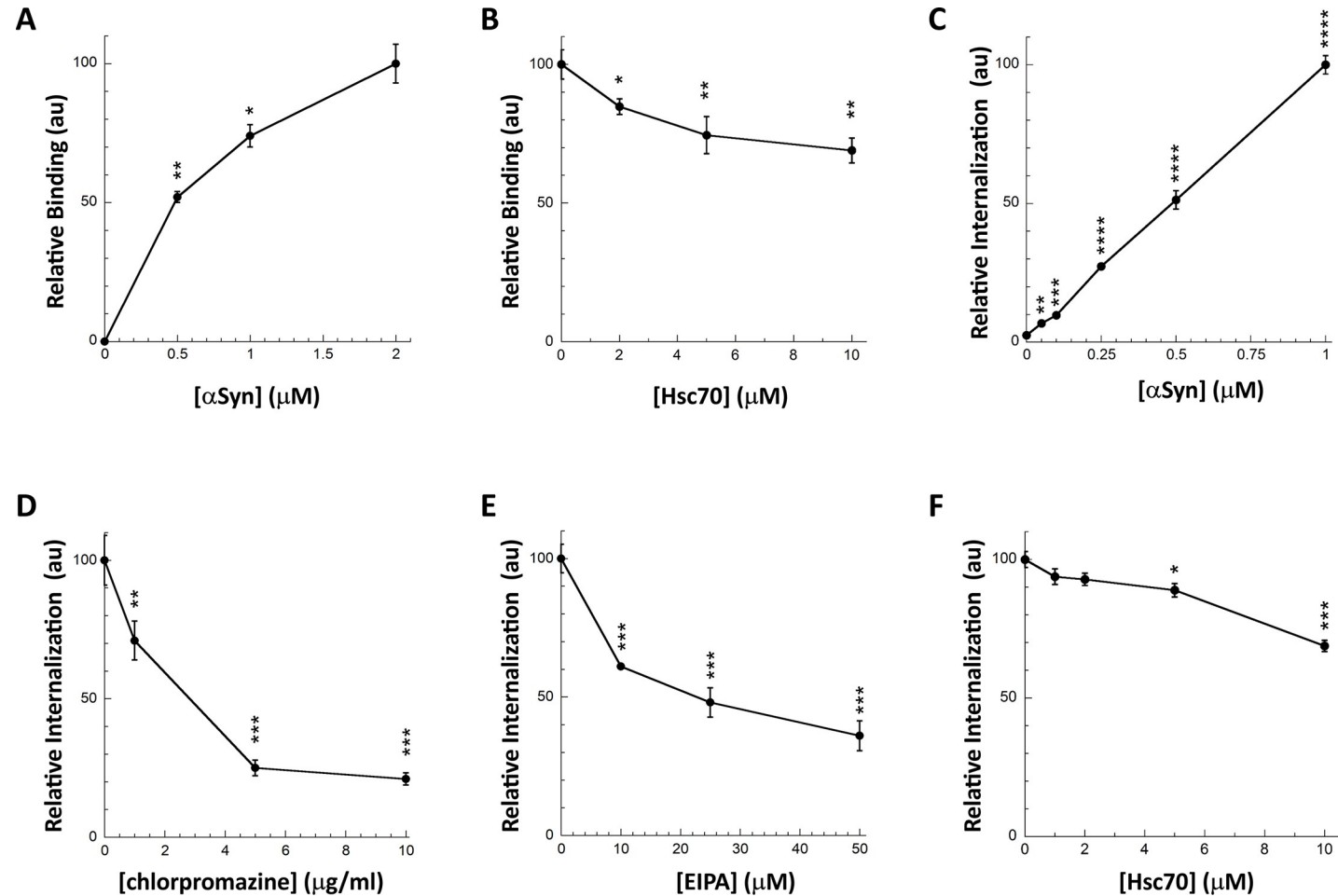

**Fig 2. Hsc70 binding to αSyn fibrils interferes with their interaction with the plasma membrane and their subsequent internalization. A,** Dose-dependent binding of αSyn fibrils to the plasma membrane of Neuro-2a cells. Neuro-2a cells were exposed for 30 min to αSyn-Alexa488 fibrils (0–2 μM). The cells were extensively washed and the fluorescence quantified. Representative images are shown in **S2A Fig**. For each concentration the mean percentage of Neuro-2a cells bound with at least 1 αSyn-Alexa488 fibrils foci and its associated standard error value was calculated from 3 independent experiments. The results and the associated significances are expressed relative to maximum binding. B, Hsc70 prevents αSyn fibrils binding to the plasma membrane. αSyn-Alexa488 fibrils (1 μM) were incubated with increasing concentrations of Hsc70 (0–10 μM) in DMEM for 30 min at 37˚C. Neuro-2a cells were next exposed to the mixture for 30 min. Fluorescence was quantified after extensive washing. Representative images are shown in **S2B Fig**. For each Hsc70 concentration, the mean percentage of Neuro-2a cells with at least one αSyn-Alexa488 fibrils foci and its associated standard error value was calculated from 3 to 5 independent experiments. The results and the associated significances are expressed relative to fibrils binding in the absence of Hsc70. **C,** αSyn fibrils take-up by Neuro-2a cells. Neuro-2a cells in 96-wells plates were exposed for 4 hours to increasing concentrations of αSyn-Alexa488 fibrils. Trypan blue was added after extensive washing to quench the fluorescence of plasma membrane-bound αSyn fibrils. The amount of internalized αSyn-Alexa488 was measured on a fluorescence plate reader. Means and their associated standard error values were calculated from 5 independent wells. The results are expressed relative to maximum internalization (1 μM αSyn fibrils). Significances are calculated in comparison to the absence of internalization (no αSyn). **D,E,** Chlorpromazine (D) and l'5-N-ethyl-isopropyl-amiloride (EIPA; E) prevent αSyn fibrils internalization by Neuro-2a cells. Neuro-2a cells in 96-wells plates were exposed for 1 hour to increasing concentrations of chlorpromazine (0–10 μg/ml) or EIPA (0–50 μM) before addition of αSyn-Alexa488 fibrils (0.5 μM). After 4 hours of incubation and extensive washing, Trypan blue was added to quench the fluorescence of plasma membrane-bound αSyn fibrils. The amount of internalized αSyn-Alexa488 was measured on a fluorescence plate reader. Means and their associated standard error values were calculated from 5 independent wells. The results and the associated significances are expressed relative to the absence of inhibitors. **F,** Hsc70 prevents αSyn fibrils internalization by Neuro-2a cells. αSyn-Alexa488 fibrils (0.5 μM) were incubated with increasing concentrations of Hsc70 (0–10 μM) in DMEM for 30 min at 37˚C. Neuro-2a cells in 96-wells plates were exposed for 4 hours to the mixture. Trypan blue was added after extensive washing to quench the fluorescence of plasma membrane-bound αSyn fibrils. The amount of internalized αSyn-Alexa488 was measured on a fluorescence plate reader. Means and their associated standard error values were calculated from 3 independent experiments. The results and the associated significances are expressed relative to internalization in the absence of Hsc70.

cultured cells were then incubated for 30 min with this mix. The data presented in **Figs 2B and S2B** clearly demonstrate that Hsc70 affects αSyn fibrils binding to Neuro-2a cells in a dose-dependent manner.

Fibrillar αSyn uptake by cells can be assessed quantitatively by fluorescence microscopy after quenching of the fluorescence at cells plasma membrane by Trypan blue (**S3 Fig**). To increase statistical power we set-up a robust 96-wells plate assay [42]. Neuro-2a cells, in 96-wells plate, were exposed for 4h to Alexa488-labeled αSyn fibrils pre-incubated with Hsc70 or not, prior to Trypan blue addition and quantification of Alexa488 fluorescence in a plate-reader. The amount of internalized fibrils was determined in a dose- (**Fig 2C**) and time-dependent manner. Fibrillar αSyn take-up was inhibited in a dose-dependent manner by chlorpromazine and l'5-N-ethyl-isopropyl-amiloride (EIPA) that inhibits clathrin-mediated endocytosis [43] and macropinocytosis [44], respectively, suggesting that the fibrils are taken up by endocytosis (**Fig 2D and 2E**). We and others previously demonstrated that endocytically internalized αSyn fibrils are then able to escape the endocytic pathway and reach the cytosol by endosomal rupture [45–47]. Preincubation of Alexa488-labeled αSyn fibrils with Hsc70 significantly affected their take-up (**Fig 2F**). We used Hoechst staining to ascertain that the number of cells remained constant (see Material & Methods).

We conclude from these observations that Hsc70 binding to αSyn fibrils affects their binding and take-up by neuronal Neuro-2a cells in a dose-dependent manner. The use of full-length Hsc70 for therapeutic purposes has drawbacks because of its pleiotropic effects within cells. We thus aimed at generating fragments of Hsc70 that retain αSyn fibrils binding capacity.

## Hsc70 Substrate Binding Domain and sub-domains retain αSyn fibrils binding capacity

In a first step toward the design of Hsc70-derived peptides that would potentially retain their ability to bind αSyn fibrils, we assessed the affinity of different Hsc70 sub-domains for αSyn fibrils (**Figs 3 and 4**). Hsc70 is composed of two domains, a Nucleotide Binding Domain (NBD), responsible for the chaperone ATPase activity, and a Substrate Binding Domain (SBD), that binds Hsc70 clients. We previously used lysine-reactive chemical cross-linkers and mass-spectrometry to map the surface areas within Hsc70 that interact with monomeric αSyn; all the identified areas were within the SBD (**Fig 3A**) [37,38]. To determine whether Hsc70 SBD retains the ability to bind αSyn fibrils we expressed and purified it. Hsc70 SBD (**Fig 3B, left**) can be subdivided in 2 sub-domains, a β-strands/sheet rich (SBDβ; **Fig 3B, middle**) and an α-helical domain, named "SBD-lid" (**Fig 3B, right**). Lysine residues from both of these sub-domains are located within the Hsc70-αSyn interaction interface suggesting that they both contribute to αSyn binding. We therefore expressed and purified SBDβ and SBD-lid.

The secondary structure content of Hsc70 SBD, SBDβ and SBD-lid was assessed by circular dichroism measurements. The data suggest that the polypeptide conformation is retained (**Figs 3B**, **4A** and S4**A Fig**). We next assessed SBD, SBDβ and SBD-lid binding to αSyn fibrils as described for full-length Hsc70 and derived dissociation constants from those measurements (**Figs 4B** and **S5 Fig**). All three domains bind αSyn fibrils. Moreover, the $K_D$ were similar to that we determined for full-length Hsc70 (**Fig 4B**). Thus, both Hsc70 SBDβ and SBD-lid contribute to fibrillar αSyn binding as for monomeric αSyn [37,38].

## αSyn fibrils Hsc70-derived peptides binders

To identify peptides derived from Hsc70 that have all that is necessary and sufficient to bind αSyn fibrils, we synthesized ten 11 to 24 residues Hsc70-derived polypeptides (**Table 1; Fig 3C**) based on the regions that contribute to αSyn binding and/or participate to the canonical substrate groove (**Fig 3A**). Some peptides were overlapping to maximise their chances of adopting the right conformation and binding (i.e. Hsc-2 and Hsc-3). Hsc-1, 2, 3, 9 and 10

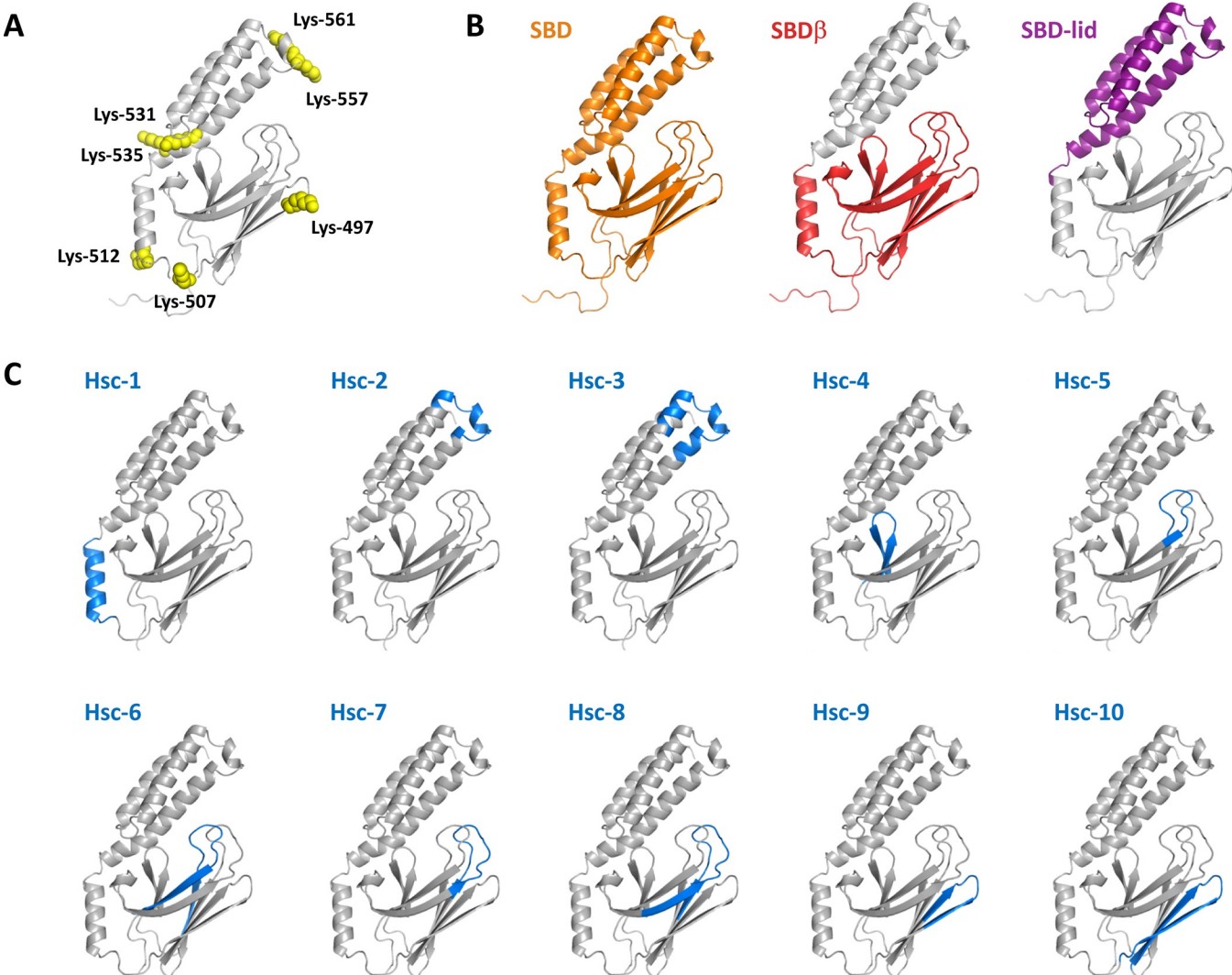

**Fig 3. Hsc70 domains and peptides used throughout this study. A**, αSyn-binding sites on Hsc70. The binding sites were determined by cross-linking Hsc70 to monomeric αSyn with chemical cross-linkers and identifying the surface interfaces by mass-spectrometry [37,38]. Only the substrate-binding domain (SBD) of Hsc70 is shown. Cross-linked lysines are depicted in yellow (space fill). Hsc70 model was built as described in [37]. **B,** Hsc70 SBD sub-domains. SBD β-sandwich (SBDβ) and lid (SBD-Lid) sub-domain are coloured. **C,** Hsc70-derived peptides. 10 peptides, which primary structure is given in Table 1, reproducing Hsc70 amino acid stretches involved in αSyn binding and the canonical Hsc70 client proteins binding sites [48], were synthesized.

encompass Hsc70-αSyn interaction surface interfaces [37,38]. Hsc-4, 5 and 6 reproduce Hsc70 canonical client binding cavity [40]. Hsc-7 and 8 decal the rest of Hsc70 SBDβ loops. Hsc-4 and 9 were found insoluble in PBS. Their interaction with αSyn was not further studied. The secondary structure content of the 8 remaining peptides was assessed by circular dichroism measurements (**Table 1** and **S4B Fig**). The peptides were predominantly unstructured, with the exception of Hsc-1 and 10 (52 and 46% α-helical, respectively). The presence of an α-helical conformation in the Hsc-1 peptide is coherent with the structure of this peptide within Hsc70 while Hsc-10 was expected to adopt a hairpin structure (**Fig 3C**).

Hsc70 binding to monomeric αSyn affects assembly into fibrils [36]. We therefore first assessed Hsc70-derived peptide capacity to interact with monomeric αSyn through their ability to affect assembly into fibrils (**Fig 5**). Monomeric αSyn assembly into fibrils was monitored

**A**

| Hsc70 domains | Hsc70 sequence | Length (aa) | Measured secondary structure composition (predicted values in brackets) | | |
|---|---|---|---|---|---|
| | | | α-helix | β-strand | Other |
| **SBD** | 382 - 646 | 265 | 28 (30) % | 26 (31) % | 46 (39) % |
| **SBDβ** | 382 - 526 | 144 | 5 (9) % | 41 (39) % | 54 (52) % |
| **SBD-lid** | 524 - 646 | 123 | 51 (52) % | 11 (0) % | 38 (48) % |

**B**

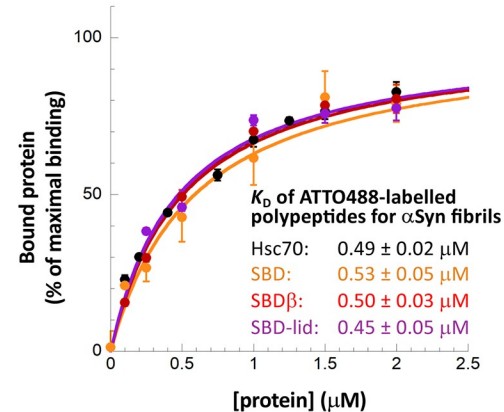

$K_D$ of ATTO488-labelled polypeptides for αSyn fibrils

Hsc70: 0.49 ± 0.02 μM
SBD: 0.53 ± 0.05 μM
SBDβ: 0.50 ± 0.03 μM
SBD-lid: 0.45 ± 0.05 μM

**Fig 4. Hsc70 domains and their binding to αSyn fibrils. A,** Secondary structure of Hsc70 SBD and sub-domains. The CD spectra used for deconvolution are shown in S4A Fig. **B,** Determination of Hsc70 domains– αSyn interactions $K_D$. The experiments were performed as in Fig 1B. In each case the normalized mean amount of labelled Hsc70 domain bound to the fibrils (% of maximal binding) and the associated standard error values were calculated from 2 to 3 independent experiments. Representative raw data are shown in S5 Fig.

using Thioflavin T (ThT) binding at 37°C in the absence or the presence of equimolar amounts of each peptide. Hsc-6 significantly slowed down αSyn assembly into fibrils while Hsc-10 accelerated aggregation (**Fig 5A and 5B**). The fibrillar nature of the assemblies obtained at the end of the reactions were assessed by transmission electron microscopy (**Fig 5C**). We conclude from these observations that 2 out of the 8 Hsc70-derived peptides we tested (Hsc-6 and 10) interact with monomeric αSyn in such a way that the time course of assembly into fibrils is significantly affected.

We next assessed Hsc70-derived peptides binding to fibrillar αSyn. Fibrillar αSyn (100 μM) was incubated with each peptide (200 μM). The fibrils were sedimented and resuspended and the bound Hsc70-derived peptides were quantified by reversed phase chromatography. The results are presented in **Table 2** and **S6A–S6F Fig**. Hsc-1, 2, 3, 5, 7 and 8, did not bind to αSyn fibrils. Hsc-6 and 10 did bind to the fibrils. As a positive control we used the aromatic molecule Surfen, which is known to bind to the SEVI amyloid fibrils and to prevent their interaction with cells [49]. Surfen was found to bind to αSyn fibrils (**Table 2**).

The affinities of Hsc-6 and 10 for αSyn fibrils were determined and the $K_D$ were over 100 μM (**S6G and S6H Fig**). Nonetheless, to determine whether Hsc70-derived peptides affect fibrillar αSyn uptake by cells, Alexa488-labeled αSyn fibrils were pre-incubated with up to 10

**Table 1. Hsc70-derived peptides primary and secondary structures.** The CD spectra used for deconvolution are shown in S4B Fig.

| Peptide | Hsc70 sequence | Sequence | Secondary structure composition α-helix β-strand other | | |
|---|---|---|---|---|---|
| **Hsc-1** | 510–525 | LSKEDIERMVQEAEKY | 52% | 0% | 48% |
| **Hsc-2** | 553–566 | VEDEKLQGKINDED | 6% | 13% | 81% |
| **Hsc-3** | 548–571 | NMKATVEDEKLQGKINDEDKQKIL | 9% | 7% | 84% |
| **Hsc-4** | 400–415 | SLGIETAGGVMTVLIK | | | |
| **Hsc-5** | 428–439 | FTTYSDNQPGV | 0% | 15% | 85% |
| **Hsc-6** | 422–444 | TKQTQTFTTYSDNQPGVLIQVYE | 0% | 20% | 80% |
| **Hsc-7** | 461–475 | LTGIPPAPRGVPQIE | 10% | 17% | 73% |
| **Hsc-8** | 457–477 | GKFELTGIPPAPRGVPQIEVT | 7% | 24% | 69% |
| **Hsc-9** | 489–500 | SAVDKSTGKENK | | | |
| **Hsc-10** | 484–505 | GILNVSAVDKSTGKENKITITN | 46% | 13% | 41% |

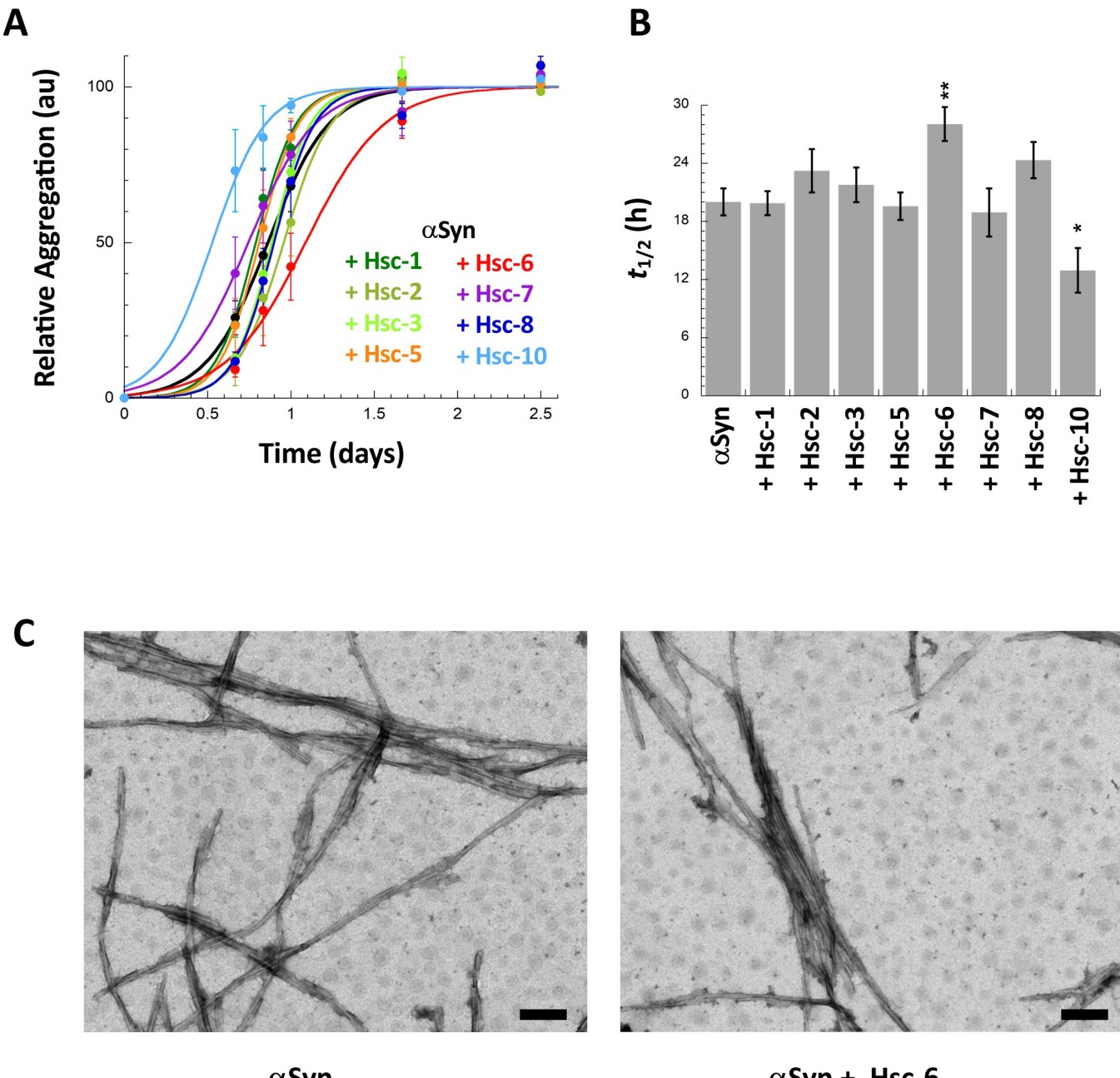

**Fig 5. Effect of Hsc70 SBD-derived peptides on αSyn aggregation. A,** Time-course of αSyn aggregation in the absence or presence of Hsc70 SBD-derived peptides. Soluble αSyn (50 μM) was incubated with or without Hsc70-derived peptides (50 μM) at 37°C and 600 rpm in PBS. The assembly reactions were monitored by Thioflavin T binding. Means and their associated standard errors values were calculated from 4 independent experiments. The lines through the data points represent the best fits to a sigmoid function. **B,** Effect of Hsc70-derived peptides on the half-time ($t_{1/2}$) of αSyn aggregation. For each independent experiment, the $t_{1/2}$ parameter was extracted from the best fit to a sigmoid function. The means and their associated standard error values were calculated from 4 independent experiments. **C,** Negative stained electron micrographs of αSyn assembled alone (left) or in the presence of equimolar concentration of Hsc-6 (right). Scale bar, 200 nm.

molar excess of the different Hsc70-derived peptides and fibrils uptake by Neuro-2a cells was quantified. None of the Hsc70-derived peptides had an effect on αSyn fibrils take-up (**Fig 6**). This is consistent with the poor affinity of the best fibrillar αSyn peptide binders. In contrast,

**Table 2. Binding of the Hsc70-derived peptides to αSyn fibrils assessed by phase reverse chromatography analysis.**

| Sample | Bound μM | peptides (%) |
|---|---|---|
| Hsc-1 | 1.2 | 0.6% |
| Hsc-2 | 1.0 | 0.5% |
| Hsc-3 | 1.2 | 0.6% |
| Hsc-5 | 2.8 | 1.4% |
| Hsc-6 | 68 | 34% |
| Hsc-7 | 1.2 | 0.6% |
| Hsc-8 | 1.6 | 0.8% |
| Hsc-10 | 72 | 36% |
| Surfen | 176 | 88% |

preincubation of αSyn fibrils with Surfen affected, in a dose-dependent way, their take-up by Neuro-2a cells (**Fig 6**).

## Peptide derived from an αSyn fibrils membrane partner, the α3 subunit of the Na$^+$/K$^+$-ATPase (NKA)

We previously brought evidence for interaction of fibrillar αSyn with the α3-subunit of the membrane ion pump NKA by pull-down [25]. α3NKA amino acid stretch that interacts with αSyn was identified by cross-linking and mass spectrometry. It consists of the extracellular loop connecting the transmembrane helices 7 and 8 [25]. Interaction of fibrillar αSyn with this extracellular loop of α3NKA was further confirmed by mutagenesis studies [25]. To determine whether NKA derived peptides affect αSyn fibrils binding to and take-up by Neuro-2a cells, we synthesized a 27 amino acid residues long peptide (NKApep) that reproduces this loop within α3NKA (**Fig 7A**). NKApep is soluble in PBS; it is disordered with some β-strand content, as assessed by circular dichroism (**Figs 7B** and S4**C**).

NKApep neither affected αSyn aggregation (not shown) nor bound to preformed αSyn fibrils (**S6D and S6F Fig**). Preincubation of preformed αSyn fibrils with NKApep resulted in a dose-dependent reduction in fibrillar αSyn binding to Neuro-2a cells (**Figs 7C** and **S2C**) but did not affect fibrils take-up (**Fig 7D**). Altogether, although designed to affect fibrillar αSyn binding to cells, NKApep acts somewhat differently, possibly through interactions with other membranous components.

## Discussion

αSyn fibrils are able to spread from one neuronal cell to another [2–4]. This process is believed to contribute to the spatiotemporal progression of pathology in the central nervous system [2,5]. The binding of αSyn fibrils to naïve cells, after their formation and release from affected counterparts, is key and has been actively documented as it constitutes a potential target for therapeutic interventions [7,13]. We hypothesized that ligands that change the surface properties of αSyn fibrils should affect binding to cell membranes. We previously showed that Hsc70 binding to αSyn fibrils affects the viability of cultured cells of neuronal origin [36]. We demonstrate here that Hsc70 interaction with αSyn fibrils compromises their binding and take-up by cells. The pleiotropic functions of full-length Hsc70 limit its therapeutic potential [50,51]. We therefore generated polypeptides reproducing Hsc70 sub-domains and surfaces that we previously showed to interact with αSyn through cross-linking studies [37,38] and assessed their effect on αSyn fibrils binding to and take-up by cells of neuronal nature. We show here that

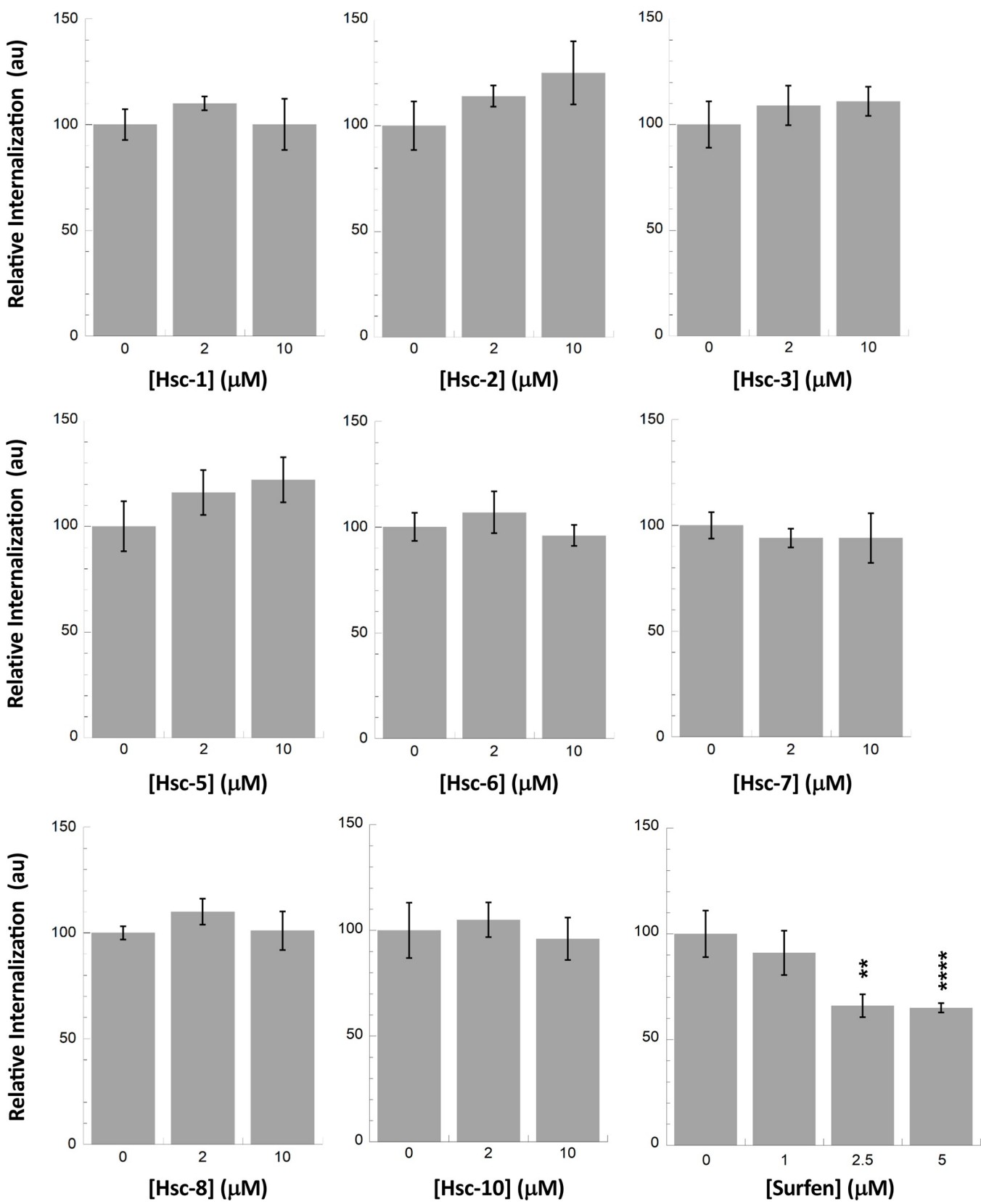

**Fig 6. Effect of Hsc70 SBD-derived peptides on αSyn fibrils take-up by Neuro-2a cells.** Each Hsc70 SBD-derived peptide (2 or 10 μM) was incubated with Alexa488-labeled αSyn fibrils (1 μM) in DMEM for 30 min at 37°C. Neuro-2a cells grown in 96-wells plates were exposed to the mixture for 2 hours. After extensive washing trypan blue was added to quench the fluorescence of plasma membrane-bound αSyn fibrils. The amount of internalized αSyn-Alexa488 was measured using a fluorescence plate reader. Means and their associated standard error values were calculated from 3 independent experiments. The same experiment was carried out with Surfen.

two peptides derived from Hsc70 SBD interact with αSyn without affecting, most probably because of their limited affinity, their take-up by Neuro-2a cells [52,53].

We previously identified through unbiased pull-down and cross-linking strategies a fibrillar αSyn neuron membrane proteins interactome [25]. Polypeptides reproducing αSyn protein partners may interfere with fibrils binding to their targets. We therefore assessed the shielding propensity of NKApep, an NKA-derived peptide that encompasses a region we showed to interact with fibrillar αSyn [25]. NKApep did not bind to αSyn fibrils under our experimental conditions, nonetheless, we show here that it interferes with fibrillar αSyn binding to cells. This suggests that NKApep affects αSyn fibrils binding to the cell indirectly, possibly through the redistribution of other αSyn fibrils target proteins [21].

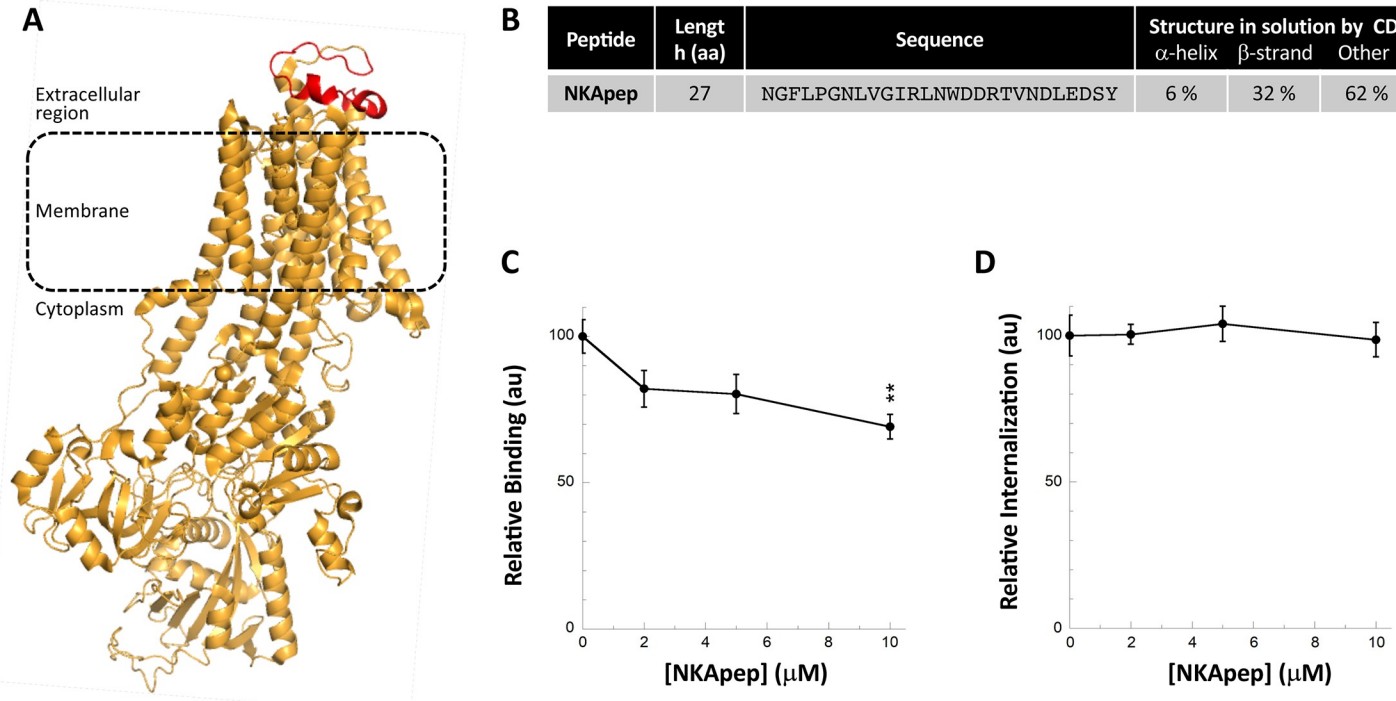

**Fig 7. Effect of a peptide derived from the α3 subunit of the NKA on αSyn fibrils binding and take-up by Neuro-2a cells. A,** Structure of the α3 subunit of the NKA *Bos taurus* (PDB 4xe5) where the 27 amino acid residues long peptide NKApep corresponding to the extracellular loop previously shown to interact with αSyn fibrils [25] is coloured in red. **B**, Secondary structure content of NKApep determined by circular dichroism. The CD spectra used for deconvolution is shown in S4C Fig. **C,** Effect of NKApep on αSyn fibrils binding to the plasma membrane of Neuro-2a cells. αSyn-Alexa488 fibrils (1 μM) were incubated without or with increasing concentrations of NKApep in DMEM for 30 min at 37°C. Neuro-2a cells were next exposed to the mixture for 30 min. Fluorescence was quantified after extensive washing. Representative images are shown in **S2B Fig**. For each peptide concentration, the mean percentage of Neuro-2a cells with at least 1 αSyn-Alexa488 fibrils foci and its associated standard error value was calculated from 3 independent experiments. The results and the associated significances are expressed relative to fibrils binding in the absence of peptide. **D,** Effect of NKApep on αSyn fibrils take-up by Neuro-2a cells. αSyn-Alexa488 fibrils (0.5 μM) were incubated with increasing concentrations of NKApep (0–10 μM) in DMEM for 30 min at 37°C. Neuro-2a cells grown in 96-wells plates were exposed to the mixture for 4 hours. After extensive washing trypan blue was added to quench the fluorescence of plasma membrane-bound αSyn fibrils. The amount of internalized αSyn-Alexa488 was measured using a fluorescence plate reader. Means and their associated standard error values were calculated from 4 independent experiments. The results are expressed relative to the internalization in the absence of peptide.

Overall, our results suggest that polypeptides that bind αSyn fibrils must have a very high affinity to affect fibrils uptake by cells and hold therapeutic potential. Advantageously, the affinity of polypeptides is amenable to improvements. They can be trimmed and modified by replacing a number of amino acid residues and reassessing affinity in an iterative manner [54]. To limit their folding landscape, they can be stapled using unnatural amino acids bearing alkyl tethers of various lengths at either one or two helix turns [55] or compatible with covalent crosslinking via click chemistry [56], or fused to a scaffolding protein such as thioredoxin [57]. Alternatively, their avidity could be increased by generating tandem repeats of the same or different peptides that bind αSyn fibrils. Many other modifications can be made so that pharmacokinetic properties of polypeptides that interfere with αSyn fibril binding and take-up by cells are improved. Thus, such peptide could yield new therapeutic tools to slow down the progression of synucleinopathies and other neurodegenerative diseases.

## Materials and methods

### Expression and purification of αSyn, Hsc70 and Hsc70 subdomains

Recombinant human wild-type αSyn was purified as described [58]. Recombinant $His_6$-tagged Hsc70 was purified as described [36]. The activity of the purified Hsc70 was assessed using a luciferase refolding assay, as described in [36].

Genes encoding the Hsc70 domain and subdomains SBD, SBDβ and SBD-lid were amplified from the pPRO-EXHTb (Invitrogen) Hsc70 vector [36] and inserted into a pET-M11 vector with an N-terminal 6xHis tag followed by a tobacco etch virus (TEV) protease cleavage site. Recombinant His-tagged proteins were expressed at 37˚C in E.coli strain BL21(DE3) (Stratagene, Santa Clara, CA) and purified as follow. Cells were harvested and resuspended in buffer A (30 mM HEPES pH 7.5, 300 mM NaCl, 10% glycerol, 20 mM imidazole). After sonication and centrifugation, lysate supernatants were filtered and loaded onto a 5 mL Talon metal affinity resin column (Clontech, Saint-Germain-en-Laye, France), equilibrated in buffer A. His tagged proteins were eluted with buffer A supplemented with 500 mM imidazole, and then dialysed in PBS. The His tags were cleaved with addition of His-TEV protease, produced using the plasmid pRK1043 (Addgene, Cambridge, MA), at a 1:25 His-TEV:chaperone molar ratio. 100% cleavage, as assessed by SDS-PAGE, was achieved upon incubating the mixtures for 1h at 37˚C. The untagged proteins were purified by collecting the flow through of a 5 mL HisTrap FF column.

Proteins concentrations were determined spectrophotometrically using the following extinction coefficients at 280 nm ($M^{-1}.cm^{-1}$): 5960 for αSyn; 39310 for Hsc70; 12950 for SBD; 2980 for SBDβ; and 9970 for SBD-lid. Pure proteins in PBS were filtered through sterile 0.22-μm filters, aliquoted and stored at -80˚C.

### Synthetic peptides and Surfen

All the peptides we designed were purchased from GL Biochem Ltd. (Shanghai, China). Peptides were dissolved in PBS at 0.5 mM, aliquoted, and stored at -20˚C. Surfen (S6951) was purchased from Sigma, dissolved in DMSO at 5 mM, aliquoted, and stored at -20˚C.

### Circular dichroism

Far-UV CD spectra were recorded at 20˚C using a JASCO J-810 dichrograph equipped with a thermostated cell holder using a 0.01-cm path length quartz cuvette. Each spectrum was the average of 10 acquisitions recorded in the 260–195 nm range with 0.5-nm steps, a bandwidth

of 2 nm, and at a speed of 50 nm/min. All spectra were buffer corrected. The spectra were deconvoluted with the Dichroweb software [59].

## Assembly of αSyn into fibrils and labelling

For fibril formation, αSyn was incubated at 200 μM in PBS at 37°C under continuous shaking in an Eppendorf Thermomixer set at 600 rpm for 2 weeks to allow completion of the aggregation reaction. The completion of the aggregation reaction was monitored by withdrawing an aliquot (100 μL), subjecting it to centrifugation in a 5415R tabletop centrifuge (Eppendorf) at 20,000g and 20°C for 30 min and assessing spectrophotometrically the amount of αSyn remaining in the supernatant. The proportion of soluble αSyn was systematically less than 10% (**S1A Fig**). The fibrillar nature of the aggregates obtained at the end of the aggregation reaction (S1**B Fig**) was assessed using a Jeol 1400 transmission electron microscope (Jeol Ltd.) following adsorption of the samples onto carbon-coated 200-mesh grids and negative staining with 1% uranyl acetate. The images were recorded with a Gatan Orius CCD camera (Gatan).

For cellular binding and internalization experiments, the fibrils were labeled by addition of the aminoreactive fluorescent dye Alexa 488 (Invitrogen, Carlsbad, CA, USA) using a protein: dye molar ratio of 10:1 based on initial monomer concentration. Labelling was performed following the manufacturer's recommendation. The reaction was stopped by adding Tris-HCl pH 7.5 (20 mM final concentration). Finally, the fibrils were sonicated with an ultrasound sonicator (Hielscher Ultrasonic, Teltow, Germany) set at an amplitude of 75 and 0.5 s cycles for 1 min.

## Binding of Hsc70, SBD, SBDβ and SBD-lid to preformed αSyn fibrils and $K_D$ determination

For binding assay, αSyn fibrils (100 μM) alone, Hsc70 alone (10 μM) or αSyn fibrils and Hsc70 (100 and 10 μM, respectively) were incubated for 1h at RT in PBS. Samples were spun for 30 min at 50,000g and 20°C in a TL100 tabletop ultracentrifuge (Beckman) and the proportion of Hsc70 present in the pellet *vs* the supernatant was analysed by SDS-PAGE.

The $K_D$ for Hsc70, SBD, SBDβ and SBD-lid interaction with αSyn fibrils were measured as follow. Hsc70 and its subdomains were first labeled by addition of the aminoreactive fluorescent dye ATTO488 (Invitrogen, Carlsbad, CA, USA) using a protein:dye molar ratio of 1:5. The reaction was stopped by adding Tris-HCl pH 7.5 (20 mM final concentration). The unreacted fluorophore was removed by NAP5 desalting column. Under these conditions the majority of primary amines unaffected by the labelling as 0.05 to 0.08 dye molecules were incorporated on average within Hsc70 or its subdomains, as assessed by absorbance spectroscopy. Binding of ATTO488-labeled polypeptides to fibrillar αSyn was then followed by a filter retardation assay where fibrils and associated proteins are retained on a membrane [60]. The different ATTO488-labeled polypeptides were diluted with their unlabeled counterpart (labeled:unlabeled polypeptides ratio of 1:50) at different final concentrations (0–2 μM) and incubated with or without αSyn fibrils (1 μM) in PBS for 1h at RT. 200 μl of each sample were filtered in triplicate through cellulose acetate membranes (0.2 μm pore size, Millipore Corp., Bedford, MA) using a 48-slot slot-blot filtration apparatus (GE Healthcare). The amount of labeled polypeptide retained on the membrane was visualized using a ChemiDocTM MP (BioRad). Images were processed and quantified using Image Lab.

Alternatively, to ensure that the labelling did not affect the binding properties of Hsc70 to αSyn fibrils, a fixed concentration of Hsc70-ATTO488 (0.2 μM) was incubated with increasing concentrations of unlabeled Hsc70 (0–10 μM) and with or without αSyn fibrils (1 μM) in PBS for 1h at RT. The experiment was then performed as above.

## Assessment of synthetic peptides effect on αSyn assembly

αSyn (50 μM monomer concentration) was incubated in the absence or in the presence of peptides (50 μM) in PBS at 37˚C under continuous shaking in an Eppendorf Thermomixer set at 600 rpm. Aliquots (10 μL) were withdrawn at different time intervals from the assembly reaction and mixed to a Thioflavin T solution (10 μM; 400 μL). Thioflavin T fluorescence was recorded with a Cary Eclipse spectrofluorimeter (Varian Medical Systems Inc.) using excitation and emission wavelengths set at 440 and 480 nm, respectively. The nature of the fibrils obtained at the end of the aggregation reaction was assessed by electron microscopy as described above. The proportion of αSyn assembled into fibrils was assessed by ultracentifugation in a TL100 tabletop centrifuge (Beckman) at 50,000g and 20˚C for 30 min and analyse of the supernatant and pellet fractions by SDS-PAGE. Following Coomassie staining / destaining the gels were visualized using a ChemiDocTM MP (BioRad). Images were processed and quantified using Image Lab.

## Binding of peptides derived from Hsc70 and NKA and Surfen to preformed αSyn fibrils and $K_D$ determination

Hsc70-derived peptides, the NKApep peptide or the Surfen molecule (0 or 200 μM) were incubated with or without αSyn fibrils (100 μM) for 1h at RT in PBS. The samples were centrifuged for 30 min in a 5415R tabletop centrifuge (Eppendorf) at 20,000g and 20˚C. The pellets were first washed by 100 μL of 0.1% TFA and then dissolved for 30 min in 30 μL of pure TFA. After TFA evaporation, the samples were resuspended in 0.1% TFA and stored at -20˚C. The composition of each sample was assessed by phase reverse chromatography on a C18 column (Jupiter C18 300A from Phenomenex, Torrance, CA, USA). The solvent composition was 0.1% TFA for solvent A and 80% acetonitrile, 0.09% TFA for solvent B, and the flow was set at 200 μl/min. The column was equilibrated in 5% B. The peptides were eluted by a gradient from 5% to 80% of solvent B. The amount of αSyn-associated ligand present in each sample was determined by comparing their respective absorbance at 215 nm (peptides) or 260 nm (Surfen) to the absorbance of a known amount of the same ligand. For $K_D$ measurements the same experiment was performed using a range of peptide concentrations (0–200 μM).

## Cell culture

Murine neuroblastoma Neuro-2a cells (ATCC, Manassas, VA) were culture at 37˚C in humidified air with 5% CO2 in Dulbecco's modified Eagle's medium (DMEM) containing 10% foetal bovine serum, 2 mM glutamine, 100 units.ml$^{-1}$ penicillin and 100 μg.ml$^{-1}$ streptomycin. All materials used for cell culture were from PAA Laboratories GmbH (Pasching, Austria).

## Binding of αSyn fibrils to Neuro-2a cells

Alexa488-labeled αSyn fibrils (1 μM equivalent monomer concentration) were first incubated for 30 min at 37˚C in DMEM without or with the ligands (Hsc70 or the NKApep peptide) at different concentrations. Neuro-2a cells cultured on ibidi-μ-Dishes (ibidi, Martinsried, Germany) were then incubated for 30 min with this mix. Then, the cells were washed and immediately imaged in serum-free, phenol red-free DMEM on a Zeiss Axio Observer Z1 epifluorescence microscope equipped with an Incubator XLmulti S2 RED LS (Carl Zeiss) and an Orca-R2 camera (Hamamatsu) at a 20x magnification. The percentage of cells with bound Alexa488 foci was estimated by randomly counting at least 500 cells in 10–15 fields and the experiments were reproduced independently 3 times. For each field the number of foci was automatically assessed using the software Fiji [61,62] and an in-house built plugin.

## Internalization of αSyn fibrils by Neuro-2a cells

Neuro-2a cells cultured on ibidi-μ-Dishes (ibidi, Martinsried, Germany) were exposed for 4h to Alexa488-labeled αSyn fibrils (1 μM equivalent monomer concentration) at 37˚C in DMEM. The cells were washed twice with serum-free, phenol red-free DMEM and 0.1% Trypan Blue (Sigma-Aldrich) was added to quench Alexa488 fluorescence at the plasma membrane. The cells were then imaged and the percentage of cells with internalized Alexa488 foci was estimated as described above.

The uptake of Alexa488-labeled αSyn fibrils (0.5 μM equivalent monomer concentration) pre-incubated or not for 30 min at 37˚C in DMEM with the ligands (Hsc70, peptides or Surfen) at different concentrations to Neuro-2A cells was also assessed using a 96-well plate assay. The cells cultured on 96-wells plates were incubated with the fibrils, preincubated or not with the ligands for 30 min at 37˚C in DMEM, in 5 independent wells. After 4 hours the media was removed and Hoechst (Sigma-Aldrich) diluted at 0.2 μg/ml in serum-free, phenol red-free DMEM was added for 30 min. The cells were washed twice with serum-free, phenol red-free DMEM and 0.1% Trypan Blue (Sigma-Aldrich) was added to quench the fluorescence of plasma membrane-bound Alexa488-labeled αSyn fibrils. For each wells Alexa488 and Hoechst fluorescences were recorded on a Clariostar microplate reader (BMG LABTECH GmbH, Germany). For each condition Alexa488 fluorescence value was considered and averaged over the 5 wells only if the Hoescht value was not significantly different from the one of untreated cells.

To assess to role of endocytosis in αSyn fibrils internalization, Neuro-2a cells cultured on 96-wells plates were first incubated with increasing concentrations of chlorpromazine or l'5-N-ethyl-isopropyl-amiloride (EIPA). After 1 hour Alexa488-labeled αSyn fibrils (0.5 μM equivalent monomer concentration) was added. The experiment was then performed as above.

## Statistical significance

Statistical significance was determined through an unpaired student's t-test. Annotations used throughout the manuscript to indicate level of significance are as follows: * $p < 0.05$; ** $p < 0.01$; *** $p < 0.001$; **** $p < 0.0001$.

## Supporting information

**S1 Fig. αSyn fibrils used for the binding and internalization studies.** Monomeric αSyn was assembled for 2 weeks at 200 μM (equivalent monomer concentration), labelled with Alexa488 and sonicated for 1 min, as described in Material & Methods. **A**, The completion of the aggregation reaction was assessed by measuring the concentration of αSyn in the supernatant at t = 0 and t = 2 weeks. The mean and associated standard deviation values were calculated from 5 independent experiments. **B**. The fibrillar nature of the resulting aggregates assessed by transmission electron microscopy after negative staining. Scale bar, 200 nm.
(PDF)

**S2 Fig. Representative epifluorescence and phase contrast images for the binding of αSyn fibrils to Neuro-2a cells. A**, Dose-dependent binding of αSyn fibrils to the plasma membrane of Neuro-2a cells. Neuro-2a cells were imaged after exposure for 30 min to αSyn-Alexa488 fibrils (0–2 μM equivalent monomer concentration) and extensive washing. **B,** αSyn fibrils binding to the plasma membrane of Neuro-2a cells in the presence or the absence of Hsp70 and NKApep. αSyn-Alexa488 fibrils (1 μM equivalent monomer concentration) were incubated in the absence (top pannels) or in the presence of Hsc70 (10 μM; middle panels) or NKApep (10 μM; bottom panels) in DMEM for 30 min at 37˚C. Neuro-2a cells were imaged

after exposure to the mixture for 30 min and extensive washing. Scale bars, 20 μM.
(PDF)

**S3 Fig. Internalization of αSyn fibrils assessed by fluorescence microscopy.** Neuro-2a cells were exposed for 4 hours to αSyn-Alexa488 fibrils (1 μM equivalent monomer concentration). The cells were washed twice with serum-free, phenol red-free DMEM then 0.1% Trypan Blue was added to quench the fluorescence of plasma membrane-bound Alexa488-labeled αSyn fibrils. Scale bars, 20 μM.
(PDF)

**S4 Fig. CD spectra of domains and peptides used throughout this study. A**, Hsc70 domains SBD, SBDβ and SBD-lid. **B**, Hsc70 peptides. **C**, NKApep.
(PDF)

**S5 Fig.** Quantification of SBD-ATTO488 **(A),** SBDβ-ATTO488 **(B)** and SBD-lid-ATTO488 **(C)** binding to αSyn fibrils. ATTO488-labelled Hsc70 SBD domain and sub-domains were diluted with the corresponding unlabelled proteins (at a molar ratio 1:50) to different final concentrations (0–5 μM) and incubated with or without αSyn fibrils (1 μM) for 1h at RT. Each sample was then filtered in triplicate through a cellulose acetate membrane and the amount of ATTO488-labelled Hsc70 domain trapped onto the membrane was quantified. In each case a representative experiment is shown. $K_D$ values presented in Fig 4B were derived from 2 to 3 independent experiments.
(PDF)

**S6 Fig. Binding of the peptides Hsc-6, Hsc-7, Hsc-10 and NKApep to αSyn fibrils and $K_D$ determination. A-F,** αSyn fibrils alone (100 μM), αSyn fibrils (100 μM) and peptides (200 μM) (panels A-D) and Peptides alone (100 μM; panels E and F), were incubated 1h at RT. The samples were centrifuged for 30 min at 20.000g and 20˚C. The pellets were dissolved in TFA 100%. After evaporation, the samples were resuspended in TFA 0.1%, and analysed by reversed phase chromatography on a C18 column. The retention time of each peptide was determined by a separate injection of 1 nmole of the peptide and is indicated by an arrow; for Hsc-7 (B) and NKApep (D), the arrow indicating the putative position of the peptide is in dotted line since no peptide was found to be associated with the αSyn pellet. Hsc-6 and Hsc-10 co-sediment with αSyn fibrils, whereas Hsc-7 and NKApep do not. **G, H,** Determination of Hsc-6 and Hsc-10 - αSyn fibrils $K_D$. Measurements as described above were performed for increasing peptide concentrations (0–200 μM). The amount of αSyn fibrils-bound Hsc-6 and Hsc-10 is plotted against the total peptide concentration. The lines through the data points represent the best fits to a linear function and are drawn for visual guidance only.
(PDF)

**S1 Raw images.**
(PDF)

## Acknowledgments

We thank Mrs Tracy Bellande for expert technical assistance. This work was supported by the Centre National de la Recherche Scientifique, the Institut National de la Santé et de la Recherche Médicale, the Région Ile de France through DIM Cerveau et Pensée, the Institut de France-Fondation Simone et Cino Del Duca, the Fondation Pour La Recherche Médicale (contract DEQ. 20160334896), the EC Joint Programme on Neurodegenerative Diseases and Agence Nationale pour la Recherche (TransPathND, ANR-17-JPCD-0002-02 and Protest-70, ANR-17-JPCD-0005-01). This work benefited from the electron microscopy facility Imagerie-Gif.

## Author Contributions

**Conceptualization:** Elodie Monsellier, Ronald Melki.

**Data curation:** Elodie Monsellier, Maya Bendifallah, Virginie Redeker.

**Formal analysis:** Elodie Monsellier, Maya Bendifallah, Virginie Redeker, Ronald Melki.

**Funding acquisition:** Ronald Melki.

**Investigation:** Elodie Monsellier, Maya Bendifallah, Virginie Redeker, Ronald Melki.

**Methodology:** Ronald Melki.

**Project administration:** Ronald Melki.

**Resources:** Ronald Melki.

**Supervision:** Ronald Melki.

**Writing – original draft:** Elodie Monsellier, Ronald Melki.

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
