## [Decision Letter · Decision Letter 0]

28 Jan 2020

PONE-D-19-35295

Polypeptides derived from a lpha-Synuclein binding partners to prevent a lpha-Synuclein fibrils interaction with and take-up by cells

PLOS ONE

Dear Dr. Melki,

Thank you for submitting your manuscript to PLOS ONE. After careful consideration, we feel that it has merit but does not fully meet PLOS ONE’s publication criteria as it currently stands. Therefore, we invite you to submit a revised version of the manuscript that addresses the points raised during the review process.

1) Concerns of Reviewer #1: Please address the 8 points raised by this reviewer. Addressing the various points will improve the readability of the manuscript.

2) Concerns of Reviewer #2:

a) Please show that the a-syn binding peptides also have the same effect on unlabeled a-syn.

b) Demonstrate that internalized a-syn reaches compartments other than the endosomes.

c) Do the inhibitory peptides prevent the cell-to-cell spread of aggregated a-syn? This may be too much at this point to carry out. Try to address this concern.

3) Editor

a) Legends to the Supplementary figures must be supplied with the revision

b) PLOS ONE now requires that submissions reporting blots or gels include original, uncropped blot/gel image data as a supplement or in a public repository. You should provide any missing raw image data for blot/gel results when they submit their first revision. 

We would appreciate receiving your revised manuscript by Mar 13 2020 11:59PM. To enhance the reproducibility of your results, we recommend that if applicable you deposit your laboratory protocols in protocols.io, where a protocol can be assigned its own identifier (DOI) such that it can be cited independently in the future. For instructions see: http://journals.plos.org/plosone/s/submission-guidelines#loc-laboratory-protocols

We look forward to receiving your revised manuscript.

Kind regards,

Stephan N. Witt, Ph.D.

Academic Editor

PLOS ONE

Journal Requirements:

2. We note that this submission reports a functional enzymological study with kinetic and thermodynamic data.

The reporting of these data should include the temperature, pH and pressure, as well as the identity of the catalyst and its origins, the method of preparation, criteria for purity and assay conditions. We recommend that you refer to the Standards for Reporting Enzymology Data (STRENDA) of the Beilstein Institut for details regarding the adequate description of experimental conditions and reporting of enzyme activity data: https://www.beilstein-strenda-db.org/strenda/public/guidelines.xhtml.

Please note that the Beilstein Institut’s STRENDA database automatically checks manuscript data for guideline compliance, as well as making them publicly available after publication and assigning them a specific DOI number for reference and tracking purposes.

If you obtain a STRENDA Registry number (SRN) and PDF containing all your functional enzymology data, please include these as Supplementary files.

Reviewers' comments:

Reviewer's Responses to Questions

**Comments to the Author**

1. Is the manuscript technically sound, and do the data support the conclusions?

Reviewer #1: No

Reviewer #2: Yes

2. Has the statistical analysis been performed appropriately and rigorously? 

Reviewer #1: No

Reviewer #2: Yes

3. Have the authors made all data underlying the findings in their manuscript fully available?

Reviewer #1: Yes

Reviewer #2: Yes

4. Is the manuscript presented in an intelligible fashion and written in standard English?

Reviewer #1: Yes

Reviewer #2: No

5. Review Comments to the Author

Reviewer #1: The study by Monsellier E et. al. is based upon their earlier findings on interaction of Hsc70 and sodium/potassium pump NaK-ATPase with α-syn. In the present study, authors designed various peptides based upon the interaction motif of Hsc70 or NKA, and investigated their potential to bind to a-syn monomer/fibrils, and ability to inhibit fibrils to bind or internalize Neuro-2a cells. They found that two of the Hsc70 based peptides though affected α-syn assembly however could not inhibit its internalization into cells. NKA based peptide did not bind to fibrils however affected α-syn fibril binding to cells without affecting their uptake. Though its an interesting attempt to design small peptides that could affect α-syn fibril uptake, the study primarily could not produce such inhibitors, and thus its not clear about the significance of the study. In addition, I do have following major concerns that must be addressed.

(1) The kinetics of α-syn fibrillation at the α-syn concentration used for interaction assay in Figure 1A is not shown. Also its important to mention that the fibrils used for interaction study were obtained from which phase of fibril formation.

(2) Figure 1B: The protocol followed to measure binding affinity is not clear. The text (Page 5, 1st paragraph) mentions that only labelled Hsc70 was incubated however the figure legend and material and Methods mention that unlabeled Hsc 70 was also added with ratio of 1:50 (lablled:unlabeled)

(3) Figure 1B: The data on filter membrane is shown for 8 different concentrations of Hsc70 however the fitted curve shows 9 data points. Also “Y’ axis is labelled as “% maximal binding” which should be defined in the Figure Legend.

(4) Figure legends for Supplementary Figures are missing at least in the version I received for review.

(5) Figure S1A and S1B: Control showing cells in the absence of α-synuclein fibrils is missing. Also instead of “αsyn” author should mention “αsyn fibrils”. Also control showing the effect of incubation of monomeric α-synuclein with neuronal cells is missing.

(6) Figure 2D and 2E: Reference indicating chloramazine and EIPA as inhibitor of clathrin-mediated endocytosis in Neuro-2a cells should be cited in the text.

(7) Figure 3A: Various peptides, based upon the interacting region of Hsc70 to α-syn, were synthesized. It seems that some of the interacting sites, such as region 531-535 was not explored. Authors should provide a justification for selecting only few of the sites shown in Figure 3A. Similarly, what is the rationale of selecting Hsc-3 peptide.

(8) Figure S5: The chromatogram for peptide alone should also be shown as a control study.

Reviewer #2: The authors present a potentially interesting study in which they consider peptides derived from a-syn binding partners to have potential as new treatment option. However, further experiments, to support the effectiveness of such a potential therapeutic, are required

The study is mainly based on take up of labeled a-syn into N2a cells.

1. Using the most effective a-syn binding peptides of this study, can the authors show that the effect on unlabelled a-syn would be the same? (perform ICC with anti-a-syn ab on n2a cells exposed to unlabelled a-syn and peptides). It would be good to verify that the internalised Alexa-488 signal does not origin partially from Alexa-488 not bound to a-syn.

2. please demonstrate that the internalised a-syn reach other compartments than the endosomes (such as the cytoplasm)

The potential of the this therapeutic approach is based on the prion-like function of a-syn: The assumption that aggregates made of mostly a-syn spread from cell to cell and could be a cause of pathology that ultimately results in parkinson's disease.

3. Therefor it must be demonstrated that this proposed therapeutic can inhibit such a mechanism rather than just uptake. It maybe too much to ask at this stage to demonstrate this in an animal model. However, if the authors can demonstrate this using mouse primary neuronal cell culture or similar ,this would strongly suggest that this proposed therapeutic holds great promise.

6. PLOS authors have the option to publish the peer review history of their article (what does this mean?). If published, this will include your full peer review and any attached files.

Reviewer #1: No

Reviewer #2: No

---

## [Author Response · Author response to Decision Letter 0]

22 Jul 2020

Response to Reviewer #1: 

The study by Monsellier E et. al. is based upon their earlier findings on interaction of Hsc70 and sodium/potassium pump NaK-ATPase with α-syn. In the present study, authors designed various peptides based upon the interaction motif of Hsc70 or NKA, and investigated their potential to bind to a-syn monomer/fibrils, and ability to inhibit fibrils to bind or internalize Neuro-2a cells. They found that two of the Hsc70 based peptides though affected α-syn assembly however could not inhibit its internalization into cells. NKA based peptide did not bind to fibrils however affected α-syn fibril binding to cells without affecting their uptake. Though its an interesting attempt to design small peptides that could affect α-syn fibril uptake, the study primarily could not produce such inhibitors, and thus its not clear about the significance of the study. 

In addition, I do have following major concerns that must be addressed.

(1) The kinetics of α-syn fibrillation at the α-syn concentration used for interaction assay in Figure 1A is not shown. Also its important to mention that the fibrils used for interaction study were obtained from which phase of fibril formation.

We understand the reviewer concerns. We mention in the Material & Method section that the aggregation reaction for the interaction assay is performed until completion, and that the completion of the reaction is assessed by pellet/supernatant partitioning and the fibrillar nature of the resulting aggregates is confirmed by TEM. Nonetheless, we present in, the revised version of the manuscript this set of data in a new Supplementary Figure (Figure S1) and we provide additional information in the Material & Method section (p20). We would like to stress that the aggregation kinetics presented in Figure 5A clearly demonstrate that steady state, as reflected by the plateau of the ThT signal, is reached after 2.5 days e.g. way less than 2 weeks. 

(2) Figure 1B: The protocol followed to measure binding affinity is not clear. The text (Page 5, 1st paragraph) mentions that only labelled Hsc70 was incubated however the figure legend and material and Methods mention that unlabeled Hsc 70 was also added with ratio of 1:50 (lablled:unlabeled)

We apologize if we were insufficiently clear. In the revised version the Results (p5) and Material & Methods (p21) sections as well as the legend of Figure 1 (p5) all consistently indicate that the experiment was performed with a mix of labelled and unlabelled Hsc70 at a 1:50 molar ratio.

(3) Figure 1B: The data on filter membrane is shown for 8 different concentrations of Hsc70 however the fitted curve shows 9 data points. Also “Y’ axis is labelled as “% maximal binding” which should be defined in the Figure Legend.

We acknowledge that the initial version was confusing and we apologize for that. The filter-trap membrane presented in Figure 1B is a single experiment representative of the 3 different experiments performed. The Hsc70 concentration range used in these 3 experiments was not exactly the same. Indeed after a first attempt for which we had only a vague idea of the measured KD, the concentration range was adjusted for the following experiments. The same principle applies for the filter-trap membranes presented in Figure S5 (S4 in the initial version) and the corresponding curves of Figure 4B. The legends of Figures 1B (p6), 4B (p10) and S5 (p33) were modified accordingly. As requested, the Y axis labelled as “% of maximal binding” has been defined as “The mean amount of Hsc70-ATTO488 bound to the αSyn fibrils normalized to the amount of Hsc70-ATTO488 bound at the maximal concentration used” in the legend of Figure 1 (p6). The legend of Figure 4B has been completed as well (p10).

(4) Figure legends for Supplementary Figures are missing at least in the version I received for review.

We apologize for this omission, which is corrected in the revised version.

(5) Figure S1A and S1B: Control showing cells in the absence of α-synuclein fibrils is missing. Also instead of “αsyn” author should mention “αsyn fibrils”. Also control showing the effect of incubation of monomeric α-synuclein with neuronal cells is missing.

We thank the Reviewer for these suggestions. We added one raw in Figure S2 (S1 in the initial version) showing representative fluorescence and phase contrasts images of Neuro-2A cells in the absence of αSyn fibrils, and we modified the labels that now specify “αSyn fibrils”.

We characterized in the past with a panel of different and complementary techniques the effect of incubation of monomeric αSyn with different neuronal cell lines. In particular in (Pieri et al, Biophys J 2012 and Scientific reports 2016) we used the same set-up as the one used in our present manuscript for assessing the binding of different αSyn species on neuronal cells, and observed no or marginal binding with αSyn monomers, no perturbation of the cell physiology as assessed through intracellular Ca2+ monitoring and MTT measurements. Thus, we feel there is no point in performing this control again. Nonetheless, following the Reviewer suggestion we added the following sentence in the main text (p6): “We previously demonstrated that monomeric αSyn barely bind in these conditions (Pieri et al, Biophys J 2012; Scientific Report 2016).”

(6) Figure 2D and 2E: Reference indicating chloramazine and EIPA as inhibitor of clathrin-mediated endocytosis in Neuro-2a cells should be cited in the text.

We thank the Reviewer for this suggestion. We added the corresponding references in the main text (p9).

(7) Figure 3A: Various peptides, based upon the interacting region of Hsc70 to α-syn, were synthesized. It seems that some of the interacting sites, such as region 531-535 was not explored. Authors should provide a justification for selecting only few of the sites shown in Figure 3A. Similarly, what is the rationale of selecting Hsc-3 peptide.

The Hsc-3 peptide and its shorter version Hsc-2 were chosen because they encompass a previously characterized interaction surface area (Lys-561 and Lys-557). As for other potential binding sites we designed two versions of the same site, a shorter (Hsc-2) and a longer one (Hsc-3), to optimize the peptide chances to adopt the conformation it adopts in Hsc70. 

The peptides Hsc-2 to Hsc-10 were designed because in addition of encompassing previously characterized interaction areas between Hsc70 and αSyn, they are situated in regions of the Hsc70 SBD domain that forms the substrate cavity. This is not the case of the region 531-535, which faces the Nucleotide Binding Domain (NBD). This is not the case of Hsc-1 neither but because it encompasses a full α-helix we anticipated that with this peptide we could have the opportunity the study a partially structured peptide. It turned out that Hsc-1 is indeed one of the few peptides we studied that has some element of secondary structure in solution.

Following the Reviewer suggestion the text in the revised version of the manuscript has been modified to more clearly specify these points (p11).

(8) The chromatogram for peptide alone should also be shown as a control study.

 We provide the chromatograms of the Hsc- and NKA-derived peptide alone spun alone under the same conditions than with a-syn fibrils and ran on the FPLC under the same conditions in Figure S6. As the reviewer can see the peptides do not sediment by themselves. 

 

Reviewer #2: 

The authors present a potentially interesting study in which they consider peptides derived from a-syn binding partners to have potential as new treatment option. However, further experiments, to support the effectiveness of such a potential therapeutic, are required

The study is mainly based on take up of labeled a-syn into N2a cells.

1. Using the most effective a-syn binding peptides of this study, can the authors show that the effect on unlabelled a-syn would be the same? (perform ICC with anti-a-syn ab on n2a cells exposed to unlabelled a-syn and peptides). It would be good to verify that the internalised Alexa-488 signal does not origin partially from Alexa-488 not bound to a-syn

We understand the reviewer concerns. We would like first to stress that there is no free Alexa-488 in our experiments. Indeed, as indicated in the material section of the revised version of the manuscript, the fibrils are spun and resuspended twice after labeling to remove the free dye (page 21). Despite this, in the case free Alexa-488 is internalized, it would not yield a punctate signal as in the figures we present. We quantified punctate fluorescence that is typical of fibrils. We would like further to stress that the dye plays a limited role, given that 1) we previously demonstrated that labelled and unlabelled aSyn fibrils bind to neuronal cells with exactly the same affinity (Monsellier et al, Scientific Rep 2016); 2) we also assessed the internalization of labelled aSyn fibrils by fluorescence microscopy (Figure S3 in the revised version) and see clear puncta with no sign of the diffuse fluorescence that internalized free Alexa488 would give. 

2. please demonstrate that the internalised a-syn reach other compartments than the endosomes (such as the cytoplasm)

We (Flavin et al, Acta Neuropathol 2017) and others (Freeman et al, PloS One 2013; Jiang et al, Scientific Reports 2017) extensively demonstrated and characterized the escape from the endosomes of αSyn fibrils in previously published studies. Albeit this is not an issue we deal with in the present work, we added the following sentence in the Results section (p8): “We and others previously demonstrated that endocytically internalized αSyn fibrils are then able to escape the endocytic pathway and reach the cytosol by endosomal rupture (Freeman et al, 2013; Jiang et al, 2017; Flavin et al, 2017)” to satisfy the reviewer.

The potential of the this therapeutic approach is based on the prion-like function of a-syn: The assumption that aggregates made of mostly a-syn spread from cell to cell and could be a cause of pathology that ultimately results in parkinson's disease.

3. Therefor it must be demonstrated that this proposed therapeutic can inhibit such a mechanism rather than just uptake. It maybe too much to ask at this stage to demonstrate this in an animal model. However, if the authors can demonstrate this using mouse primary neuronal cell culture or similar ,this would strongly suggest that this proposed therapeutic holds great promise.

We understand the reviewer point of view. We and other brought ample evidences for prion-like propagation if fibrillar a-syn. We think that primary neurons (we used in the past) are neither better nor worst/more or less relevant than the cell line we used. Primary neurons as the reviewer certainly know are juvenile cells that are not representative of a mature neuron. We therefore see no point in repeating such costly and dispensable measurements using primary neurons in the present COVID-19 situation. By the way, we would have to select a primary neuronal population as we recently showed that different neurons do not bind equally well fibrillar a-syn (Courtes J et al. Si Rep 2020). We would like to add that it is premature to consider in vivo experiments before a thorough improvement of the affinity of the peptides for fibrillar a-syn via an iterative mutagenesis study.

---

## [Decision Letter · Decision Letter 1]

24 Jul 2020

Polypeptides derived from alpha-Synuclein binding partners to prevent alpha-Synuclein fibrils interaction with and take-up by cells

PONE-D-19-35295R1

Dear Dr. Melki,

We’re pleased to inform you that your manuscript has been judged scientifically suitable for publication and will be formally accepted for publication once it meets all outstanding technical requirements.

Kind regards,

Stephan N. Witt, Ph.D.

Academic Editor

PLOS ONE

Additional Editor Comments (optional):

Reviewers' comments:

Reviewer's Responses to Questions

**Comments to the Author**

1. If the authors have adequately addressed your comments raised in a previous round of review and you feel that this manuscript is now acceptable for publication, you may indicate that here to bypass the “Comments to the Author” section, enter your conflict of interest statement in the “Confidential to Editor” section, and submit your "Accept" recommendation.

Reviewer #1: All comments have been addressed

2. Is the manuscript technically sound, and do the data support the conclusions?

Reviewer #1: Yes

3. Has the statistical analysis been performed appropriately and rigorously? 

Reviewer #1: Yes

4. Have the authors made all data underlying the findings in their manuscript fully available?

Reviewer #1: Yes

5. Is the manuscript presented in an intelligible fashion and written in standard English?

Reviewer #1: Yes

6. Review Comments to the Author

Reviewer #1: Authors have satisfactorily responded to the comments raised. I congratulate authors for the nice study.

7. PLOS authors have the option to publish the peer review history of their article (what does this mean?). If published, this will include your full peer review and any attached files.

Reviewer #1: No

---

## [Editor Report · Acceptance letter]

30 Jul 2020

PONE-D-19-35295R1 

Polypeptides derived from α-Synuclein binding partners to prevent α-Synuclein fibrils interaction with and take-up by cells 

Dear Dr. Melki:

I'm pleased to inform you that your manuscript has been deemed suitable for publication in PLOS ONE. Congratulations! Your manuscript is now with our production department. 

Kind regards, 

on behalf of

Dr. Stephan N. Witt 

Academic Editor

PLOS ONE